# Lysine or Threonine Deficiency Decreases Body Weight Gain in Growing Rats despite an Increase in Food Intake without Increasing Energy Expenditure in Response to FGF21

**DOI:** 10.3390/nu15010197

**Published:** 2022-12-30

**Authors:** Joanna Moro, Gaëtan Roisné-Hamelin, Catherine Chaumontet, Patrick C. Even, Anne Blais, Celine Cansell, Julien Piedcoq, Claire Gaudichon, Daniel Tomé, Dalila Azzout-Marniche

**Affiliations:** AgroParisTech, UMR PNCA, Institut National de la Recherche pour l’Agriculture, l’Alimentation et l’Environnement (INRAE), Université Paris-Saclay, 91120 Palaiseau, France

**Keywords:** lysine deficiency, threonine deficiency, energy metabolism, FGF21, energy expenditure, food intake

## Abstract

The objective of this study is to evaluate the effects of a strictly essential amino acid (lysine or threonine; EAA) deficiency on energy metabolism in growing rats. Rats were fed for three weeks severely (15% and 25% of recommendation), moderately (40% and 60%), and adequate (75% and 100%) lysine or threonine-deficient diets. Food intake and body weight were measured daily and indirect calorimetry was performed the week three. At the end of the experimentation, body composition, gene expression, and biochemical analysis were performed. Lysine and threonine deficiency induced a lower body weight gain and an increase in relative food intake. Lysine or threonine deficiency induced liver FGF21 synthesis and plasma release. However, no changes in energy expenditure were observed for lysine deficiency, unlike threonine deficiency, which leads to a decrease in total and resting energy expenditure. Interestingly, threonine severe deficiency, but not lysine deficiency, increase orexigenic and decreases anorexigenic hypothalamic neuropeptides expression, which could explain the higher food intake. Our results show that the deficiency in one EAA, induces a decrease in body weight gain, despite an increased relative food intake, without any increase in energy expenditure despite an induction of FGF21.

## 1. Introduction

Supplying protein and essential amino acids (EAA) by diet is mandatory for survival. A diet deficient in protein does not support protein synthesis and growth, leading to lower lean mass. However, protein deficiency induced an increase in food and energy intake, interpreted as an attempt to increase protein intake, but with a risk of energy overfeeding and increased adiposity [1,2,3,4,5]. However, it was observed in some studies that this was also associated with increased energy expenditure, which could be the signal responsible for the increased energy intake, and could compensate for energy overfeeding and lowering the increase in fat mass and adiposity [1,2,3,4,6,7]. In the context of protein transition toward plant protein sources, the lower protein content and poor-quality plant protein sources, arise the question of the metabolic impact of protein sources deficient in EAA on energy metabolism, mainly lysine and threonine, the first limiting EAA in human nutrition.

Low EAA supply is considered the main effector of protein deficiency. Indeed, a diet low in all EAA (but adequate in all non-essential amino acids) was able to reproduce the effects of a protein-deficient diet on body weight, food intake, and energy expenditure [2,6,8]. A diet deficient in only one EAA has been shown to alter growth [9,10,11,12,13,14,15,16,17,18,19,20,21], increase or decrease food intake [6,9,10,11,12,13,15,16,18,21], modify body composition [9,10,12,13,15,16,17,18,21,22], and change or not energy expenditure [6,18,22]. However, the implication of one or several specific amino acids (AA) during protein deficiency was mainly investigated with a restricted number of EAA and few levels of deficiency. Indeed, a majority of studies reported the effects of only one level (for lysine [6,20,21,21,23,24,25,26,27,28,29,30], threonine [6,20,21,31], or methionine [13,14,15,16,18,21,32,33]) or two levels (for lysine [34]) of deficiencies. Indeed, whether the type of EAA or the level of deficiency was involved in these effects remains to be clarified.

Fibroblast growth factor 21 (FGF-21) is one major key hormone that regulates food intake and metabolic adaptation in response to the diet’s protein content. Low EAA supply induced liver expression and plasma release of FGF-21 that in turn modulates the metabolic and feeding response to protein and EAA deficiency [2,8,35]. Both EAA and non-EAA have been reported to induce FGF21. Depletion of hepatic and plasma glutamine [36], or alanine [37], leads to the activation of FGF21. Five NEAA (glutamate, aspartate, glutamine, asparagine, and/or alanine) were described to be sufficient to attenuate or prevent the induction by NEAA deficiency of FGF-21 mRNA expression in the liver in both FAO cells and primary mouse hepatocytes [38]. Among EAAs, Maida et al. 2016 reported that a restriction of phenylalanine and methionine had no effect [38], whereas others reported that methionine is sufficient to increase FGF21 [15,18,39,40,41]. For BCAA, some reported no effect of BCAA [42] or leucine [38,42]. Others have found that leucine restriction increased the production of FGF21 [43,44]. Threonine and tryptophan deficiencies have been also described as required to confer the response to EAA restriction, mediated by FGF21 [45], whereas lysine deficiency seems not required [6,45]. FGF21 in turn increased energy expenditure only for threonine and tryptophan deficiency [6]. Taken together, these studies showed conflicting data, only a few levels of EAA deficiency were studied, and the role of AA deficiency in the induction of FGF21 remains to be clarified.

In this context, this study aims to assess the effects of different levels of EAA deficiencies (lysine or threonine) on food and energy intake, body weight, body composition, energy metabolism, and expression of FGF-21 in growing rats. The novelty of this study was to investigate the effect of these EAA at different levels of deficiency, i.e., severely (15% and 25% of recommendation), moderately (40% and 60%), and adequate (75% and 100%). Moreover, we compared the effect of two strictly EAA, to investigate the specific effect of each of these EAA deficiencies to understand the metabolic impact of plant protein sources deficient in EAA.

## 2. Materials and Methods

### 2.1. Animals

A total of one hundred and twelve male Wistar Han rats (3 weeks old, weighing 50–55 g on their arrival; HsdHan: WISt^®^, Envigo, France) were acclimated to the light and temperature-controlled facility of AgroParisTech (12 h/12 h reversed light/dark cycle, light on from 00:00 to 12:00, temperature 22 ± 1 °C) for one week, before the experiment. During this adaptation period, rats were fed a standard rat chow diet (“A04” from Safe, 19.30% of protein). Studies were approved by the Regional Animal Care and Ethical Committee and Minister of Research and conformed to the European legislation on the use of laboratory animals (registration number: APAFIS #13436-2017122616504600).

### 2.2. Experimental Diets

The composition of experimental diets is shown in Table A1 and Table A2 (Appendix A). Two types of diet were formulated at 20% energy equivalent protein (with a mixture of total milk protein and free amino acid) varying either in lysine or in threonine contents only.

Diets were designed in reference to lysine or threonine content of 9.2 or 6.2 g·kg_diet_^−1^, corresponding to the estimated requirements for growth [46]. These requirements are covered by a diet containing 20% energy as highly digestible protein and balanced amino acid composition diet, as in the AIN-93G diet, which is considered to meet protein and EAA requirements in growing rats [46]. For each diet, the desired quantity of lysine or threonine was provided by a corresponding amount of total milk protein, and the diet was completed for all other amino acids in a free crystalline form to the equivalent amount of the 20% energy total milk protein diet. Diets were referenced as L or T, related to the % of requirement provided by the diet for lysine (L) or threonine (T). Diets were classified, according to the level of deficiency of lysine and threonine, as severely deficient (15% and 25% of recommendation), moderately deficient (40% and 60% of recommendation), and adequate (75% and 100% of recommendation). Free alanine was added to obtain a similar amount of total amino acids among diets. Thus, all diets were isonitrogenous and isocaloric. Two control diets were used, a 20% total milk proteins diet (P20, corresponding to the recommendation) and a control of the “diet formulation” effect, with 3% total milk proteins, completed with free amino acids including lysine and threonine (L/T100) to be equivalent to 20% total milk proteins.

### 2.3. Experimental Design

A total of one hundred and twelve rats were divided into two main groups (lysine or threonine) in this study, with fifty-six rats in each group. After a one-week adaptation to laboratory conditions, the fifty-six rats from each main group were randomly divided into seven subgroups (n = 8/group). These subgroups were assigned for three weeks to one of the 7 isocaloric diets. At the end of the 3 week experimental period, rats were fed with a calibrated meal (4 g, 58.2 kJ) of their experimental diet, and they were anesthetized two hours later, with isoflurane.

### 2.4. Food Intake, Body Weight and Body Composition Measurements

On each day, a calibrated meal of 4 g (58.2 kJ) was given at 12:00 a.m. (onset of night period), to train rats to quickly ingest this meal. Then, ad libitum access to food was given between 12:30 am and 9:00 a.m. the next day. Body weight and food intake were measured every morning for 3 weeks. At the end of the experiment, rats were euthanized, and body composition was analyzed by dissection and weighing main tissues and organs.

### 2.5. Energy Expenditure Measurement by Indirect Calorimetry

During the third week, each rat was placed for 48 h in a cage connected to an indirect calorimeter) [47], to measure energy expenditure and spontaneous motor activity as previously described [4]. Among the 2 days of measurements, the first day was devoted to habituation and the second day to data analysis. Oxygen consumption and carbon dioxide production were measured in each cage for 2 min every 10 min (2 min by cage and 2 min on room air to correct values for room O_2_% and CO_2_%). From O_2_ and CO_2_ values, the modified Weir formula [48] (to obtain data in J·s^−1^, i.e., Watt) was used to estimate total metabolic rate (TMR), follows: TMR(W) = [16.3 × VO_2_ (mL·min^−1^) + 4.7 × VCO_2_ (mL·min^−1^)]/60. Spontaneous motor activity was calculated as raw activity data cleared from background noise and multiplied by a diffusion coefficient (to maximize the correlation between spontaneous motor activity and TMR). Cost of activity was determined as the slope of the linear regression between spontaneous motor activity and TMR. Resting metabolic rate (RMR) was then obtained as follows: RMR (W) = TMR (W) − [Activity (Arbitrary Unit (AU)) × Cost of activity (AU·W^−1^)]. Thus, activity metabolic rate (AMR) was determined as: AMR(W) = TMR(W) − RMR(W). Then, to compare groups by taking into account the differences in body weight and body composition, TMR, RMR, and AMR were normalized to the rat metabolic active mass (MAM) (defined as lean body mass + 20% of fat mass). Finally, values were converted in kJ, resulting in total energy expenditure (TEE), resting energy expenditure (REE), and activity energy expenditure (EE-act).

### 2.6. Biochemical Analysis and mRNA Expression Measurements

At the end of the experimentation, blood samples were taken from the vena cava. After centrifugation (4 °C, 3000 rpm, 10 min), plasma was collected, aliquoted, and stored at −80 °C until analysis. Pieces of liver, gastrocnemius muscle, hypothalamus, and epididymal adipose tissue (EAT) were frozen in liquid nitrogen (−80 °C) for further measurement of mRNA abundance and biochemical analysis.

Plasma FGF-21 was determined using an enzyme-linked immunoassay (mouse and rat FGF-21 ELISA, BioVendor, Karasek, Czech Republic). For mRNA expression measurements, total RNAs were extracted from liver, gastrocnemius muscle, epididymal adipose tissue (EAT), and hypothalamus using Trizol reagent (Invitrogen, Waltham, MA, USA). After extraction, RNA concentration was measured using a spectrophotometer at 260 nm (Nanodrop, Waltham, MA, USA). Retro-transcription was performed on 0.4 µg of RNA using the High-Capacity cDNA Archive Kit (Applied Biosystems, Waltham, MA, USA). Then, to measure gene expression, real-time PCR was performed using Power SYBR Green PCR Master Mix (Applied Biosystems) on Step One (Applied Biosystems, Waltham, MA, USA) on 5 ng of cDNA. Finally, gene expression was calculated as 2^−ΔCt^, where ΔCt = Ct_gene_ − Ct_housekeeping gene_, and the housekeeping genes used were 18S (for liver, muscle, and EAT) or RPL13A (for hypothalamus). To detect any potential contamination, negative controls, without enzyme or RNA, were performed. Primer Express was used to design the primer sequences of genes. The sequences of used primers are described in Table A3 (Appendix B).

### 2.7. Statistical Analysis

Data are presented as the means + standard error of the mean (SEM). Statistical analysis was performed using R^©^ version 4.0.2 with R^©^ studio version 1.3.959. Repeated measures models were used to analyze the evolution of body weight during the experiment. Final models were retained depending on their Akaike Information Criterion and Bayesian Information Criterion between four variance estimation structures (compound symmetry, heterogeneous compound symmetry, self-regressive and auto-regressive heterogeneous). Post hoc analyses were performed with Bonferroni correction. One-way ANOVA was performed to test diet effect on final body weight, food intake, body composition, energy expenditure, plasma FGF-21 levels, and gene expressions. Gaussian distribution, homoscedasticity, and lack of abnormal values were the hypothesis of our statistical model, and were visually checked before analysis. If Gaussian distribution was not confirmed, the model was log-transformed to best fit a normal distribution. To conclude on inter-group differences, post hoc analysis was done by Tukey-HSD method with Bonferroni correction. The results of inter-groups are presented by letters. Data that do not share the same letter are statistically different at a risk of α = 5%.

## 3. Results

### 3.1. Food Intake, Body Weight and Food Efficiency

Lysine diets increased absolute daily energy intake (g or kJ) for L25, L60, and L100 groups (*p* < 0.001), and threonine diets for severe deficient groups (T15 and T25) (*p* < 0.001) (Table 1 and Table 2). When expressed as relative to the metabolic active mass (MAM), for lysine and threonine, severe deficient groups (L/T15 and L/T25), and L/T40 increased their relative energy intake (*p* < 0.001) (Table 1 and Table 2, Figure 1). The relative daily energy intake was also slightly higher (22%) for L/T100 comparatively to the P20 diet (*p* < 0.001), whereas it was higher by 50 to 70% for the severe deficient groups.

Lysine severe deficient groups (L15 and L25) have a significantly reduced BW gain from day 3 and 6, respectively (*p* < 0.001) (Figure 1). Under lysine diets, no difference was significant between the moderately deficient (L40 and L60) or adequate (L75 and L100) groups and the P20 group. Threonine severe deficient groups (T15 and T25) have a significantly reduced BW gain from day 1, and the lowest moderately deficient group (T40) from day 6 (*p* < 0.001) (Figure 1). Under threonine diets, no difference was significant between T60, adequate (T75 and T100), and P20 groups.

At the end of the experimental period, lysine and threonine severe deficient groups were lighter than the P20 group (*p* < 0.001) (Figure 1). Furthermore, for threonine, the T40 group was additionally lighter than the P20 group (*p* < 0.001) (Figure 1). For both amino acid deficiencies, no other differences were significant.

L15 and threonine severe deficient (T15 and T25) groups had a reduced food efficiency (*p* < 0.001), which became null for L15 and was more pronounced for the severe threonine deficient group (reduced to negative values) (Table 1 and Table 2).

### 3.2. Body Composition

Lysine severe deficient groups had a reduced Lean Body Mass (LBM) (*p* < 0.001) with a reduced weight of the liver (*p* < 0.001), kidneys (*p* < 0.001), gastrocnemius muscle (*p* < 0.001) and carcass weights (*p* < 0.001) (Table 3). Under lysine diets, there was only an increased absolute fat mass for L60 and L100 groups (*p* < 0.001), due to an increased epididymal (*p* < 0.001), mesenteric (*p* < 0.001), and retroperitoneal (*p* < 0.001) fats for L60 groups, and an increased retroperitoneal fat for L100 group (*p* < 0.001). When expressed as % of BW (i.e., adiposity), lysine severe deficient and moderately deficient groups increased their adiposity (*p* < 0.001). No difference in body composition was found between the L75 and the P20 group. Moreover, no lysine diet effect was observed on brown adipose tissue weight and, despite a significant diet effect (<0.01) for subcutaneous fat weight, no group was found significantly different from P20.

Under threonine diets, there was a decrease in LBM for threonine severe deficient groups (T15 and T25) and for the T40 group (*p* < 0.001), with reduced liver (*p* < 0.001), kidneys (*p* < 0.001), gastrocnemius muscle (*p* < 0.001), and carcass weights (*p* < 0.001) (Table 4). Threonine severe deficient groups decreased and the T60 group increased their absolute fat mass (*p* < 0.001). More precisely, threonine severe deficient groups have reduced epididymal (*p* < 0.001), mesenteric (*p* < 0.001), retroperitoneal (*p* < 0.001), subcutaneous fats, and brown adipose tissue (*p* < 0.001), and the T60 group had increased mesenteric (*p* < 0.001) and subcutaneous fat (*p* < 0.001). When expressed as adiposity, severe deficient diets as the T60 groups kept their significant difference (*p* < 0.001). No difference was found between the T75 and P20 groups for all organs and tissues. No differences were observed between T100 and P20 except for liver weight which is slightly lower for T100.

### 3.3. Energy Expenditure

Data of energy expenditure are shown in Figure 2. No effect of lysine deficiency was found on Resting Energy Expenditure (REE) or cost of activity, and, despite a significant diet effect, no group was different from the P20 group in terms of total energy expenditure (TEE) (*p* < 0.05) or activity energy expenditure (EE-act) (*p* < 0.01). For threonine diets, T15 and T75 groups have a reduced TEE (*p* < 0.01) and REE (*p* < 0.001). Despite a significant diet effect, no group was found significantly different from the P20 group for EE-act (*p* < 0.01) or cost of activity (*p* < 0.01) under threonine diets.

### 3.4. Expression of Genes in Liver, Adipose Tissue, Muscle and Hypothalamus

In the liver, under lysine diets (Table 5), no diet effect was observed for mRNA expression of glycolysis genes (GK and L-PK). For fatty acid (FA) oxidation, no diet effect was observed on CD36 or ACOX expression, and despite a significant diet effect on CPT1a expression (*p* < 0.05), no group was significantly different from P20. For lipogenesis, no diet effect was observed for ACCa and FAS, as for MTTP expression. Severe deficient lysine groups (L15 and L25) have a strongly higher expression of FGF21 mRNA (*p* < 0.001). No significant difference in mRNA expression was observed between moderately low and adequate groups for all genes. Thus, in the liver, a lysine deficiency does not seem to alter glucose or lipid metabolism; however, a severe lysine-deficient diet induces a strong FGF-21 production in liver.

Under threonine diets (Table 6), no diet effect was observed on liver-relative mRNA expression of glycolysis genes (GK and L-PK). For FA oxidation, no diet effect was found on CD36 but there was a higher ACOX expression (*p* < 0.001) for T75. Furthermore, the T60 group had a higher CPT1a expression (*p* < 0.01). For lipogenesis, no diet effect was found on ACCa expression, and despite a significant diet effect of FAS (*p* < 0.05), no group was significantly different from P20. Additionally, no diet effect was observed on MTTP expression. Severe deficient threonine groups and T40 have a strongly higher expression of FGF21 mRNA in the liver (*p* < 0.001), as T100 does. No other difference was found significant with the P20 group. Thus, in the liver, there was no clear effect of a threonine deficiency on glucose and lipid metabolism; however, a threonine-deficient diet induces a stronger FGF-21 production than a lysine-restricted one, and thus, from a lower deficiency.

In epididymal adipose tissue (EAT), under lysine diets (Table 5), no significant diet effect was observed on lipogenesis genes (ACCa and FAS). Furthermore, no diet effect was observed on the relative mRNA expression of FA oxidation (CD36). For genes involves in the browning of white adipose tissue, a significant diet effect was observed for UCP1 and UCP2 (*p* < 0.05 for both), without any group different from P20. No diet effect was observed on the expression of UCP3.

For threonine diets (Table 6), no diet effect was observed on relative mRNA expression of lipogenesis (ACCa and FAS) and FA oxidation (CD36) genes. Additionally, for genes involved in the browning of white adipose tissue, the T15 group had a strongly higher expression of UCP1 (*p* < 0.001). However, no significant diet effect was found on UCP2 expression, and despite a significant diet effect (*p* < 0.05), no group was different from the P20 group for UCP3 expression. Thus, in EAT, there was no alteration of lipid metabolism during a lysine or threonine deficiency, and we did not observe any activation of browning of WAT under lysine deficiency, whereas a severe threonine deficiency does.

In gastrocnemius muscle, under lysine restriction (Table 5), a significant diet effect was observed on mRNA expression of genes involved in fatty-acid oxidation (ACOX and CTP1b) (*p* < 0.01 for both). Indeed, the L15 group had a higher ACOX and CPT1b expression than P20. Despite a significant diet effect (*p* < 0.05), no group was found significantly different from P20 for CD36 expression.

For threonine diets (Table 6), no significant diet effect was observed on muscle mRNA expression of genes involved in fatty acid oxidation (ACOX and CPT1b). Moreover, despite a significant diet effect (*p* < 0.001), no group was found significantly different from P20 for CD36 expression.

We have previously reported that mRNA encoding FGF21 in the hypothalamus responds to the diet’s protein level. Thus, we investigated in the present study if a deficiency of one EAA can reproduce this effect. For the hypothalamus (Table 5), no group was found significantly different from the P20 group for FGF21 mRNA relative expression under lysine diets. Moreover, no diet effect was observed on mRNA expression of FGF receptors FGF-R1, -R2B, and R3, and despite a significant diet effect of FGF-R2C, no group was found different from P20. For threonine diets (Table 6), no diet effect was observed on FGF21 expression. Furthermore, no diet effect was either observed on the expression of FGF receptors FGF-R1, -R2B, -R2C, and R3.

To understand the modifications of energy intake induced by the deficiencies, we investigated the hypothalamic expression of neuropeptides and their receptors involved in the control of energy balance.

In the hypothalamus, under lysine deficiency (Table 5), no diet effect was observed on the expression of orexigenic neuropeptides (NPY and AGRP) or associated receptors (Y2R). For anorexigenic neuropeptides, no diet effect was observed on POMC expression as on CART expression. In the same way, there was no significant diet effect on the anorexigenic neuropeptides-associated receptors (MOR, MC3R, and MC4R). Furthermore, no significant diet effect was observed on the relative CRF expression.

Compared to lysine, severe threonine deficiencies (Table 6) induced some alteration of hypothalamic expression of neuropeptides. Indeed, severe threonine deficient groups (T15 and T25) had a higher expression of orexigenic neuropeptides (NPY and AGRP; *p* < 0.001 for both). However, no diet effect was observed on the associated receptor (Y2R). Simultaneously with the increased orexigenic neuropeptides, severe threonine-deficient diets had a decreased anorexigenic neuropeptide expression (POMC; *p* < 0.001). As for previous neuropeptides, no diet effect was observed on associated receptors (MOR, MC3R, and MC4R). Moreover, no diet effect was observed on the hypothalamic expression of CRF.

### 3.5. Plasma FGF-21

For both lysine and threonine deficiency, there was a strong increase in FGF21 plasma levels (*p* < 0.001 for both) (Figure 3). For lysine diets, compared to the P20 group, this effect was found only for severe deficient diets, and the level was about 41 and 54 times higher for L15 and L25, respectively. No other difference was found significant under lysine deficiency. However, for threonine diets, plasma levels of FGF-21 were higher for severe deficient diets, and for moderately deficient diet T40. Indeed, when compared to the P20 group, the level was about 96 and 99 times higher for T15 and T25, respectively, and about 34 times for T40. Furthermore, plasma FGF21 was slightly higher for the T60, T75, and T100 groups. However, although we observed an increased level of plasma FGF-21 for T60 and T75, it seems that it is artefact due to a very low FGF-21 level of the P20 group during the threonine study. Indeed, when we replace P20 from the threonine study with those from the lysine study, only T15, T25, T40, and T100 remain significantly different from the P20.

## 4. Discussion

The present study investigated the consequence of lysine or threonine deficiency in growing rats, two strictly essential amino acids, on food intake, body weight, body composition, energy expenditure and metabolism, FGF-21 expression, and plasma level.

In our work, a severe EAA (either lysine or threonine) deficiency caused a decrease in body weight gain. However, this reduction of body weight gain was not due to a reduced food intake as, relative to their metabolic active mass, deficient rats have an increased food and energy intake. Rats submitted to severe deficiency of either lysine or threonine showed increased relative food and energy intake but lower BW gain, leading to a null (for lysine) or negative (for threonine) food efficiency, as previously observed for lysine in gilt [9], chick [10] and mice [6], and for threonine in Pekin ducks [12] and rats [6]. However, Yap et al. 2020, reported in mice that threonine induced a food inefficiency sufficient enough to mimic the effects of dietary EAA restriction, they did not observe any effect of lysine deficiency [6]. In contrast, lysine deficiency induced both lower food intake and lower BW gain in chicks [10,11], and threonine deficiency induced a lower BW gain and absolute feed intake in Pekin ducks [12]. Moreover, as previously observed for piglets under moderately low threonine deficiency (~T70 diet) [31], and for growing pigs and adult pigs, but not for piglets, under moderately low lysine deficiency (~L70 diet) [49], we did not observe any modification of either food intake or BW gain under moderately deficiency in rats in the present study. Our results, in accordance with most studies but not with the results of Yap et al. 2020 for lysine [6], confirm that a deficiency in one EAA induces an appetite for protein or for corresponding EAA in order to compensate the deleterious effects of a deficiency.

To better understand the higher food intake induced by both EAA deficiencies, we measured the mRNA expression of hypothalamic neuropeptides and their associated receptors. The protein/EAA craving would be mediated through a reward [27]. In our study, we observed a slight regulation of hypothalamic neuropeptides by threonine deficiency. Indeed, a threonine deficiency caused a shift toward food intake, by increasing orexigenic (NPY and AGRP) and decreasing anorexigenic (POMC) neuropeptides, without altering any associated receptors. In line with this observation, a decreased POMC mRNA expression was observed under a low protein diet [4], which could be a signal for threonine deficiency. Thus, the increased food intake under threonine deficiency could be mediated by the regulation of hypothalamic neuropeptides. However, if these data clearly indicate an increased appetite for threonine, our results showed a different mechanism for lysine since lysine deficiency did not induce any modifications in hypothalamic neuropeptides. In contrast, the specific appetite for lysine was supported by the observation that an intragastric lysine load during lysine restriction was reported to stimulate the dopaminergic system in rats [28]. The difference between threonine and lysine may be explained by the more drastic effect of threonine on growth retardation as opposed to lysine. Moreover, a specific appetite for EAA would involve different mechanisms, depending on the EAA. It was reported that lysine deficiency upregulates ghrelin in the hypothalamus and downregulates it in the intestine [50], and upregulates leptin and adiponectin in the hypothalamus and liver in broiler chickens. Moreover, lysine deficiency induced transcription of NPy peptide in mandarin fish [51]. Indeed, the mechanisms involved in the control of specific lysine intake, in response to lysine deficiency, may be different than threonine deficiency and requires more investigation.

In our previous work performed on low protein diets, hypothalamic FGF-21 mRNA expression was correlated to food intake, and thus was proposed as being regulated in this manner [4]. However, in the hypothalamus, lysine and threonine deficiency did not modify FGF21 mRNA expression. Thus, changes in FGF-21 in the hypothalamus would be likely due to an adjustment of carbohydrate content, since the diet had a low proportion of protein, it is compensated with carbohydrates in order to have isoenergetic diets. In the present study, only the content in one EAA is different between diets. In other work, it has been proposed that the impact of protein restriction on feeding behavior could be mediated by hepatic FGF-21 signal to the brain through FGF21 receptors (FGFR) and β-Klotho (KLB) [2], and this pathway could be also involved in the increase in food intake induced by EAA deficiency. Thus, rather than hypothalamic FGF21, hepatic FGF21 could be involved in the control of increased food intake under EAA deficiency, which requires further investigation.

However, even if EAA deficiency induces a higher food intake, this did not allow to compensate for lower BW gain. EAA deficiency induces a decrease in metabolic active mass, due to a lower lean body mass. However, lysine and threonine severe deficiency acts differently on fat mass. Indeed, there was a decrease in fat mass and adiposity under very low threonine diets, whereas there was no difference in fat mass, or in increased adiposity, under very low lysine diets. Thus, both lysine and threonine deficiency reduced lean body mass, but adiposity was increased by lysine deficiency and decreased by threonine deficiency. It has been reported that lysine deficiency induced a reduction in breast muscle growth in chicks [10], and did not modify the fat-to-lean ratio in gilt [9]. Moreover, threonine deficiency decreased the relative weight of breast muscle and abdominal fat, without changing the relative weight of thigh muscle, and increased liver total lipid, triglycerides, and cholesterol, in Pekin ducks [12].

These changes in body composition did not lead to systemic modifications of energy expenditure. Indeed, lysine deficiency did not modify energy expenditure, whereas threonine deficiency induced a decrease in total energy expenditure (TEE) and resting energy expenditure (REE), without affecting activity energy expenditure or cost of activity. Lysine deficiency showed no change in the components of energy expenditure (TEE, REE, EE-act, and cost of activity) and this is in line with other results reporting no effect of lysine deficiency on energy expenditure [6]. For threonine deficiency, compared to the control group, both EE-act and cost of activity were not different, but TEE and REE were reduced while an increase in total metabolic rate (TMR) was previously observed [6]. However, the latter study was conducted at a temperature below thermoneutrality, which is reported to increase resting metabolic rate (RMR) by non-shivering thermogenesis, and by changing the activity pattern [52,53]. Thus, we can hypothesize that threonine deficiency reduces TEE and REE near thermoneutrality but when animals are kept at lower temperatures, TMR and RMR increased to allow a better adaptation to cold temperatures.

FGF21 is a key hormone which controls energy expenditure in response to a low protein and AA diet. In our study, the main result is that EAA deficiency alone is able to induce an important increase in liver mRNA expression and plasma levels of FGF-21. Interestingly, this increase is dose dependent for threonine deficiency whereas for lysine deficiency, FGF21 is induced for L15 and L25. This result agrees with the idea that increased plasma FGF-21 induced by a low protein or amino acid diet [1,7] can be reproduced by a deficiency in only one EAA, lysine or threonine. It was previously observed that intraperitoneal threonine or tryptophan, but not histidine or phenylalanine, was effective to reverse the increase in plasma FGF-21 but not for lysine [6]. In our study, if we showed that lysine and threonine deficiency are involved as signals for the expression and release of FGF21 by the liver, then energy expenditure did not follow the same variation since it increased for lysine and not for threonine deficiency. The energy expenditure enhancement of FGF21 was thought to be mediated mainly by brown fat thermogenesis which produces heat through the function of uncoupling protein [54]. In the present study, the unchanged energy expenditure in lysine-deficient fed rats was in accordance with the lack of difference of mRNA expression in EAT for UCP1, specific to BAT [55], UCP2 ubiquitously expressed [56] and UCP3 mainly specific to skeletal muscle [56], indicating no stimulation of browning of WAT, and no changes in brown adipose tissue. Whereas we have shown that a highly severe threonine-deficient diet stimulates the browning of WAT, as it increases UCP1 expression in EAT. Moreover, we observed a decrease in all fat mass tissues, including BAT. In addition, there was no difference in mRNA hepatic expression of genes involved in the glycolysis pathway, indicating that liver glucose metabolism was not altered by EAA deficiency. No difference was observed in FA oxidation or lipogenesis in the liver, contrasting to what we previously reported under low protein diets [4]. Thus, reduced fat mass under threonine deficiency is not due to an alteration of FA metabolism, and in this case, WAT browning could be a mechanism to keep a minimum amount of BAT to maintain thermogenesis. The reduced REE observed only under threonine deficiency would be thus due to an alteration of the thermogenesis ability of rats. In the study of Yap et al. 2020, changes in FGF21 in response to threonine deficiency were accompanied by an increase in energy expenditure, whereas no such changes were observed for lysine deficiency [6]. Interestingly, FGF21 is not always associated with an increase in energy expenditure and could increase body temperature independently of energy expenditure and UCP1 changes, by decreasing heat loss [57]. Thus, the difference between lysine and threonine effect on FGF21 and energy expenditure may involve different processes and warrants further investigation.

## 5. Conclusions

The present study focused on lysine and threonine deficiencies, which are the two limiting EAA in cereals, and the subsequent effects on energy metabolism. Unlike the majority of studies that reported the effects of only one level or two levels of deficiencies, the originality of the present work was to investigate the effect of these EAA at different levels of deficiency, i.e., severely (15% and 25% of recommendation), moderately (40% and 60%), and adequate (75% and 100%). Moreover, the second originality of this work is to compare the effect of two strictly EAA, lysine, and threonine. Our results show that the deficiency in one EAA, induces a decrease in body weight gain, despite an increased relative food intake, without any increase in energy expenditure even though the induction of FGF21. Moderate EAA deficiency had a marginal impact on energy metabolism which suggests the importance to study the phenotypical and metabolic impact at different levels of EAA deficiency. Threonine deficiency displayed a more pronounced impact on body mass and lean mass than lysine, suggesting a specific impact of each EAA on energy metabolism.

## Figures and Tables

**Figure 1 nutrients-15-00197-f001:**
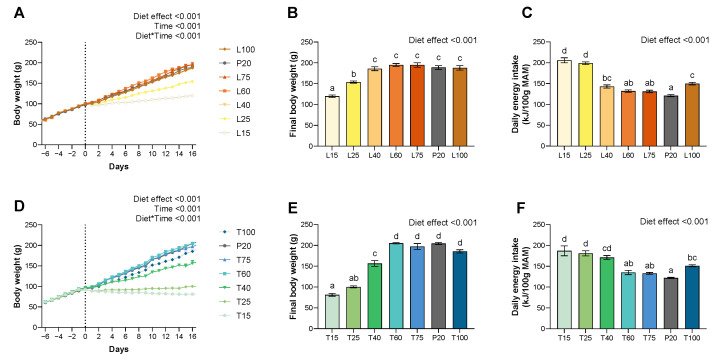
Effect of lysine or threonine deficiency on body weight gain, final body weight, and energy intake. (**A**) Evolution of body weight under lysine diets. (**B**) Final body weight for lysine diets on day 16. (**C**) Daily energy intake for lysine diets. (**D**) Evolution of body weight under threonine diets. (**E**) Final body weight for threonine diets on day 16. (**F**) Daily energy intake for threonine diets. Data that do not share the same letter are different at *p* < 0.05. Values are expressed as means + SEM.

**Figure 2 nutrients-15-00197-f002:**
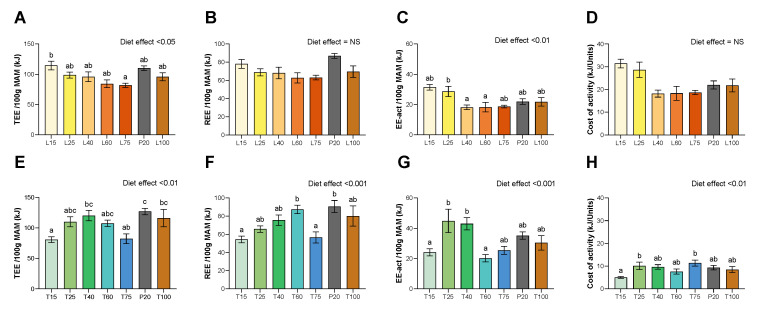
Total energy expenditure (TEE), resting energy expenditure (REE), activity energy expenditure (EE-act) and cost of activity, as obtained by indirect calorimetry. (**A**–**D**) Under lysine diets. (**E**–**H**) Under threonine diets. Values are expressed as means + SEM. Data that do not share the same letter are different at *p* < 0.05.

**Figure 3 nutrients-15-00197-f003:**
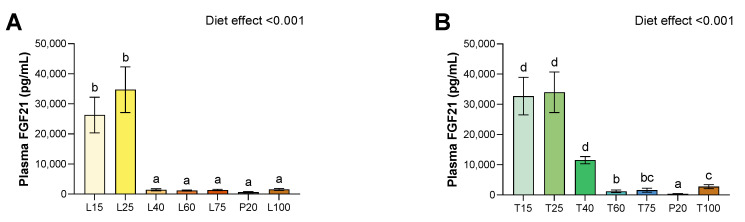
Effect of lysine or threonine deficiency on plasma FGF-21 levels. (**A**) Plasma FGF-21 levels for lysine diets. (**B**) Plasma FGF-21 levels for threonine diets. Values are expressed as means + SEM. Data that do not share the same letter are different at *p* < 0.05.

**Table 1 nutrients-15-00197-t001:** Absolute and relative values of energy and food intake for lysine diets.

Diet	L15	L25	L40	L60	L75	P20	L100	Diet Effect
Cumulated food intake (g)	285.38 ± 10.92 ^a^	**367.40 ± 4.00 ^c^**	333.33 ± 5.46 ^bc^	**341.80 ± 6.29 ^c^**	331.67 ± 11.18 ^bc^	305.03 ± 5.68 ^ab^	**361.09 ± 11.04 ^c^**	<0.001
Daily food intake (g/100 g MAM)	**14.19 ± 0.42 ^a^**	**13.70 ± 0.22 ^a^**	**9.82 ± 0.27 ^bc^**	9.09 ± 0.20 ^cd^	8.98 ± 0.21 ^cd^	8.31 ± 0.19 ^d^	**10.30 ± 0.20 ^b^**	<0.001
Daily energy intake (kJ)	244.09 ± 9.34 ^c^	**314.45 ± 3.43 ^a^**	285.29 ± 4.68 ^ab^	**292.74 ± 5.38 ^a^**	284.06 ± 9.58 ^ab^	261.42 ± 4.86 ^bc^	**308.84 ± 9.44 ^a^**	<0.001
Daily energy intake (kJ/100 g MAM)	**206.39 ± 5.96 ^a^**	**199.38 ± 3.28 ^a^**	**142.85 ± 3.96 ^bc^**	132.37 ± 2.95 ^cd^	130.72 ± 3.04 ^cd^	121.12 ± 2.78 ^d^	**149.83 ± 2.97 ^b^**	<0.001
Food efficiency (kJ/kJ)	**0.03 ± 0.01 ^c^**	0.09 ± 0.01 ^b^	0.13 ± 0.01 ^ab^	0.13 ± 0.01 ^ab^	0.12 ± 0.01 ^ab^	0.10 ± 0.01 ^ab^	0.12 ± 0.01 ^ab^	<0.001

The level of lysine deficiency is indicated by the number in diet’s name expressed in percentage of recommendation, except for control (P20). Values are expressed as means + SEM. Data that do not share the same letter are different at *p* < 0.05. Bold values are those significantly different from P20 group.

**Table 2 nutrients-15-00197-t002:** Absolute and relative values of energy and food intake for threonine diets.

Diet	T15	T25	T40	T60	T75	P20	T100	Diet Effect
Cumulated food intake (g)	**172.92 ± 9.97 ^d^**	**207.85 ± 9.40 ^d^**	335.35 ± 7.59 ^abc^	**346.69 ± 13.24 ^ab^**	333.19 ± 9.26 ^abc^	306.79 ± 5.84 ^bc^	**349.61 ± 11.26 ^a^**	<0.001
Daily food intake (g/100 g MAM)	**12.87 ± 0.82 ^a^**	**12.47 ± 0.43 ^a^**	**11.78 ± 0.32 ^ab^**	9.25 ± 0.35 ^cd^	9.12 ± 0.15 ^cd^	8.40 ± 0.10 ^d^	**10.35 ± 0.17 ^bc^**	<0.001
Daily energy intake (kJ)	**147.89 ± 8.53 ^c^**	**177.90 ± 8.04 ^c^**	289.02 ± 6.50 ^ab^	**296.93 ± 11.34 ^a^**	285.37 ± 7.93 ^ab^	262.93 ± 5.00 ^ab^	299.02 ± 9.63 ^a^	<0.001
Daily energy intake (kJ/100 g MAM)	**187.25 ± 12.00 ^a^**	**181.47 ± 6.19 ^a^**	**171.36 ± 4.67 ^ab^**	134.68 ± 5.02 ^cd^	132.84 ± 2.12 ^cd^	122.45 ± 1.46 ^d^	**150.52 ± 2.42 ^bc^**	<0.001
Food efficiency (kJ/kJ)	**−0.10 ± 0.01 ^c^**	**−0.04 ± 0.01 ^bc^**	0.08 ± 0.01 ^ab^	0.13 ± 0.01 ^a^	0.16 ± 0.01 ^a^	0.15 ± 0.01 ^a^	0.12 ± 0.00 ^a^	<0.001

The level of threonine deficiency is indicated by the number in diet’s name expressed in percentage of recommendation, except for control (P20). Values are expressed as means + SEM. Data that do not share the same letter are different at *p* < 0.05. Bold values are those significantly different from P20 group.

**Table 3 nutrients-15-00197-t003:** Absolute and relative values of body composition for lysine diets.

Diet	L15	L25	L40	L60	L75	P20	L100	Diet Effect
Initial body weight (g)	62.34 ± 0.82	61.73 ± 1.09	61.73 ± 1.09	61.00 ± 0.89	63.61 ± 0.69	63.33 ± 0.94	62.64 ± 1.14	NS
Final body weight (g)	**128.64 ± 3.36 ^d^**	**174.24 ± 3.05 ^c^**	217.63 ± 6.03 ^b^	240.88 ± 6.70 ^a^	232.13 ± 3.65 ^ab^	228.79 ± 5.90 ^ab^	223.96 ± 7.00 ^ab^	< 0.001
MAM (g)	**70.46 ± 2.03 ^d^**	**96.34 ± 1.43 ^c^**	126.53 ± 3.17 ^b^	138.88 ± 3.76 ^ab^	138.18 ± 1.81 ^ab^	135.93 ± 4.02 ^ab^	130.65 ± 3.69 ^ab^	<0.001
Fat mass (g)	13.02 ± 0.72 ^c^	20.40 ± 1.22 ^ab^	21.39 ± 1.56 ^ab^	**23.73 ± 1.30 ^a^**	18.98 ± 1.67 ^abc^	15.45 ± 1.15 ^bc^	**22.10 ± 2.29 ^a^**	<0.001
Adiposity (%)	**10.09 ± 0.40 ^bc^**	**11.66 ± 0.55 ^c^**	**9.78 ± 0.54 ^bc^**	**9.86 ± 0.47 ^bc^**	8.13 ± 0.64 ^ab^	6.42 ± 0.42 ^a^	**9.76 ± 0.80 ^bc^**	<0.001
Lean body mass (g)	**67.86 ± 1.91 ^d^**	**92.26 ± 1.30 ^c^**	122.25 ± 2.96 ^b^	134.13 ± 3.66 ^ab^	134.39 ± 1.65 ^ab^	132.84 ± 3.90 ^ab^	126.25 ± 3.33 ^ab^	<0.001
Liver (g)	**3.90 ± 0.15 ^c^**	**7.11 ± 0.37 ^c^**	7.11 ± 037 ^b^	8.88 ± 0.43 ^a^	8.73 ± 0.27 ^a^	8.34 ± 0.28 ^ab^	7.76 ± 0.40 ^ab^	<0.001
Kidneys (g)	**1.10 ± 0.04 ^d^**	**1.38 ± 0.03 ^cd^**	1.67 ± 0.06 ^ac^	1.78 ± 0.06 ^a^	1.74 ± 0.04 ^ab^	1.76 ± 0.07 ^ab^	1.46 ± 0.14 ^bc^	<0.001
Gastrocnemius muscle (g)	**1.18 ± 0.04 ^c^**	**1.52 ± 0.04 ^b^**	2.00 ± 0.04 ^a^	2.08 ± 0.09 ^a^	2.08 ± 0.05 ^a^	2.09 ± 0.07 ^a^	2.17 ± 0.14 ^a^	<0.001
Carcass (g)	**44.8 ± 1.19 ^c^**	**59.48 ± 0.91 ^b^**	77.09 ± 1.64 ^a^	83.35 ± 2.40 ^a^	84.5 ± 1.18 ^a^	84.44 ± 2.84 ^a^	78.6 ± 1.62 ^a^	<0.001
Epididymal fat (g)	1.76 ± 0.12 ^c^	2.67 ± 0.22 ^bc^	3.36 ± 0.34 ^ab^	**3.90 ± 2.29 ^a^**	3.01 ± 2.88 ^ab^	2.65 ± 0.20 ^bc^	3.25 ± 0.37 ^ab^	<0.001
Mesenteric fat (g)	1.97 ± 0.14 ^c^	2.94 ± 0.19 ^ab^	3.21 ± 0.17 ^ab^	**3.84 ± 0.22 ^a^**	3.04 ± 0.26 ^ab^	2.65 ± 0.09 ^bc^	3.36 ± 0.3 ^ab^	<0.001
Retroperitoneal fat (g)	1.88 ± 0.16 ^d^	3.35 ± 0.20 ^abc^	3.71 ± 0.30 ^abc^	**4.56 ± 0.38 ^a^**	3.10 ± 0.43 ^bcd^	2.31 ± 0.17 ^cd^	**4.15 ± 0.50 ^ab^**	<0.001
Subcutaneous fat (g)	7.41 ± 0.41 ^b^	11.44 ± 0.77 ^a^	11.10 ± 0.85 ^ab^	11.43 ± 0.78 ^a^	9.81 ± 1.07 ^ab^	7.84 ± 0.90 ^ab^	11.34 ± 1.24 ^a^	<0.01
Brown adipose tissue (g)	0.43 ± 0.03	0.53 ± 0.04	0.52 ± 0.04	0.53 ± 0.03	0.44 ± 0.03	0.44 ± 0.03	0.50 ± 0.06	NS

The level of lysine deficiency is indicated by the number in diet’s name expressed in percentage of recommendation, except control (P20). Values are expressed as means + SEM. Data that do not share the same letter are different at *p* < 0.05. Bold values are those significantly different from P20 group.

**Table 4 nutrients-15-00197-t004:** Absolute and relative values of body composition for threonine diets.

Diet	T15	T25	T40	T60	T75	P20	T100	Diet Effect
Initial body weight (g)	62.71 ± 1.79	62.44 ± 1.67	62.74 ± 1.68	63.31 ± 1.72	61.21 ± 1.56	64.95 ± 1.23	60.59 ± 1.69	NS
Final body weight (g)	**81.63 ± 1.68 ^d^**	**102.14 ± 2.16 ^d^**	**180.89 ± 4.12 ^c^**	241.79 ± 3.69 ^a^	233.59 ± 55.46 ^ab^	230.96 ± 5.91 ^ab^	215.61 ± 7.32 ^b^	<0.001
MAM (g)	**44.29 ± 1.13 ^e^**	**57.67 ± 1.27 ^d^**	**106.86 ± 2.08 ^c^**	141.28 ± 3.21 ^a^	137.71 ± 2.83 ^ab^	137.87 ± 4.06 ^ab^	126.01 ± 4.61 ^b^	<0.001
Fat mass (g)	**2.94 ± 0.26 ^d^**	**5.16 ± 0.43 ^d^**	16.48 ± 0.75 ^c^	**26.47 ± 1.45 ^a^**	23.43 ± 1.76 ^ab^	19.89 ± 1.82 ^bc^	21.05 ± 1.60 ^abc^	<0.001
Adiposity (%)	**3.60 ± 0.31 ^c^**	**5.00 ± 0.35 ^c^**	8.94 ± 0.34 ^ab^	**10.94 ± 0.57 ^a^**	10.0 ± 0.63 ^ab^	8.58 ± 0.70 ^b^	9.69 ± 0.44 ^ab^	<0.001
Lean body mass (g)	**43.70 ± 1.10 ^e^**	**56.77 ± 1.17 ^d^**	**103.63 ± 2.00 ^c^**	135.98 ± 3.23 ^a^	133.03 ± 2.74 ^ab^	133.89 ± 3.93 ^ab^	121.80 ± 4.34 ^b^	<0.001
Liver (g)	**2.77 ± 0.10 ^d^**	**3.07 ± 0.08 ^d^**	**6.11 ± 0.31 ^c^**	8.47 ± 0.20 ^a^	7.91 ± 0.25 ^ab^	8.75 ± 0.35 ^a^	**7.33 ± 0.36 ^b^**	<0.001
Kidneys (g)	**0.74 ± 0.01 ^c^**	**0.88 ± 0.02 ^c^**	**1.41 ± 0.03 ^b^**	1.70 ± 0.04 ^a^	1.69 ± 0.05 ^a^	1.78 ± 0.06 ^a^	1.65 ± 0.05 ^a^	<0.001
Gastrocnemius muscle (g)	**0.84 ± 0.02 ^c^**	**1.07 ± 0.02 ^c^**	**1.78 ± 0.05 ^b^**	2.23 ± 0.08 ^a^	2.19 ± 0.09 ^a^	2.28 ± 0.09 ^a^	2.13 ± 0.16 ^ab^	<0.001
Carcass (g)	**29.43 ± 0.71 ^d^**	**38.42 ± 0.86 ^c^**	**68.31 ± 1.26 ^b^**	84.85 ± 2.12 ^a^	82.25 ± 1.95 ^a^	84.31 ± 2.41 ^a^	77.16 ± 2.88 ^a^	<0.001
Epididymal fat (g)	**0.31 ± 0.19 ^c^**	**0.54 ± 0.05 ^c^**	2.31 ± 0.29 ^b^	4.27 ± 0.35 ^a^	3.84 ± 0.26 ^a^	3.28 ± 0.45 ^ab^	3.26 ± 0.40 ^ab^	<0.001
Mesenteric fat (g)	**0.44 ± 0.05 ^d^**	**0.76 ± 0.06 ^d^**	2.46 ± 0.14 ^c^	**4.06 ± 0.34 ^a^**	3.80 ± 0.27 ^ab^	3.31 ± 0.35 ^bc^	3.25 ± 0.23 ^bc^	<0.001
Retroperitoneal fat (g)	**0.16 ± 0.04 ^c^**	**0.39 ± 0.07 ^c^**	2.06 ± 0.15 ^b^	4.62 ± 0.31 ^a^	4.08 ± 0.38 ^a^	3.44 ± 0.49 ^ab^	3.61 ± 0.46 ^a^	<0.001
Subcutaneous fat (g)	**2.03 ± 0.19 ^c^**	**3.60 ± 0.28 ^c^**	9.35 ± 0.59 ^b^	**12.98 ± 0.68 ^a^**	11.70 ± 1.11 ^ab^	9.86 ± 0.70 ^b^	10.92 ± 0.66 ^ab^	<0.001
Brown adipose tissue (g)	**0.18 ± 0.02 ^b^**	**0.27 ± 0.03 ^b^**	0.50 ± 0.04 ^a^	0.55 ± 0.05 ^a^	0.51 ± 0.06 ^a^	0.47 ± 0.03 ^a^	0.54 ± 0.06 ^a^	<0.001

The level of threonine deficiency is indicated by the number in diet’s name expressed in percentage of recommendation, except for control (P20). Values are expressed as means + SEM. Data that do not share the same letter are different at *p* < 0.05. Bold values are those significantly different from P20 group.

**Table 5 nutrients-15-00197-t005:** Absolute relative expression of mRNA as determined by RT-qPCR for lysine diets.

Organ	Diet	L15	L25	L40	L60	L75	P20	L100	Diet Effect
Liver	GK	0.98 ± 0.30	0.89 ± 0.16	0.83 ± 0.34	0.63 ± 0.14	0.68 ± 0.13	1.00 ± 0.20	0.59 ± 0.14	NS
	L-PK	1.37 ± 0.25	1.29 ± 0.17	0.97 ± 0.17	0.99 ± 0.15	1.09 ± 0.18	1.00 ± 0.30	1.10 ± 0.14	NS
	CD36	1.20 ± 0.29	0.65 ± 0.09	0.79 ± 0.11	0.69 ± 0.06	0.69 ± 0.15	1.00 ± 0.11	0.74 ± 0.11	NS
	CPT1a	1.83 ± 0.34 ^b^	0.95 ± 0.11 ^ab^	0.94 ± 0.18 ^ab^	0.66 ± 0.06 ^a^	0.63 ± 0.10 ^a^	1.00 ± 0.13 ^ab^	0.81 ± 0.10 ^ab^	<0.05
	ACOX	1.21 ± 0.19	0.70 ± 0.06	0.73 ± 0.16	0.69 ± 0.05	0.80 ± 0.09	1.00 ± 0.13	0.82 ± 0.07	NS
	ACCa	1.28 ± 0.22	0.85 ± 0.08	0.76 ± 0.12	0.70 ± 0.08	0.72 ± 0.06	1.00 ± 0.08	0.81 ± 0.05	NS
	FAS	2.38 ± 0.54	1.56 ± 0.11	1.26 ± 0.25	1.09 ± 0.19	1.15 ± 0.18	1.00 ± 0.28	1.19 ± 0.20	NS
	MTTP	1.23 ± 0.14	0.91 ± 0.06	0.85 ± 0.12	0.81 ± 0.07	0.87 ± 0.07	1.00 ± 0.10	0.91 ± 0.05	NS
	FGF21	**11.58 ± 3.85 ^b^**	**10.06 ± 3.09 ^b^**	1.12 ± 0.37 ^a^	0.70 ± 0.16 ^a^	1.50 ± 0.32 ^a^	1.00 ± 0.27 ^a^	1.08 ± 0.28 ^a^	<0.001
Muscle	ACOX	**1.94 ± 0.19 ^b^**	1.25 ± 0.13 ^ab^	1.51 ± 0.29 ^ab^	1.43 ± 0.18 ^ab^	1.06 ± 0.15 ^a^	1.00 ± 0.13 ^a^	1.58 ± 0.20 ^ab^	<0.01
	CD36	2.59 ± 0.42 ^b^	1.90 ± 0.79 ^ab^	1.72 ± 0.41 ^ab^	1.18 ± 0.16 ^ab^	0.82 ± 0.14 ^a^	1.00 ± 0.12 ^ab^	1.35 ± 0.24 ^ab^	<0.05
	CPT1b	**2.11 ± 0.39 ^b^**	1.01 ± 0.08 ^a^	1.46 ± 0.36 ^ab^	1.23 ± 0.14 ^ab^	0.86 ± 0.12 ^a^	1.00 ± 0.13 ^a^	1.34 ± 0.20 ^ab^	<0.01
EAT	ACCa	1.08 ± 0.25	1.12 ± 0.16	0.83 ± 0.11	1.00 ± 0.05	0.55 ± 0.12	1.00 ± 0.20	0.73 ± 0.12	NS
	FAS	1.27 ± 0.21	1.32 ± 0.15	0.81 ± 0.07	1.15 ± 0.07	0.81 ± 0.20	1.00 ± 0.19	0.86 ± 0.14	NS
	CD36	1.17 ± 0.22	1.00 ± 0.11	0.85 ± 0.16	0.76 ± 0.06	0.81 ± 0.16	1.00 ± 0.16	0.77 ± 0.15	NS
	UCP1	4.28 ± 1.33 ^b^	2.77 ± 1.27 ^ab^	0.72 ± 0.21 ^a^	1.22 ± 0.49 ^ab^	1.28 ± 0.74 ^ab^	1.00 ± 0.25 ^ab^	1.06 ± 0.31 ^ab^	<0.05
	UCP2	0.97 ± 0.16 ^ab^	1.08 ± 0.08 ^ab^	1.17 ± 0.20 ^ab^	0.99 ± 0.08 ^ab^	0.78 ± 0.13 ^a^	1.00 ± 0.06 ^ab^	0.94 ± 0.08 ^a^	<0.05
	UCP3	1.20 ± 0.22	0.91 ± 0.18	0.79 ± 0.15	1.06 ± 0.15	0.83 ± 0.11	1.00 ± 0.16	0.88 ± 0.20	NS
Hypothalamus	FGF21	1.54 ± 0.44	0.81 ± 0.15	0.73 ± 0.14	0.69 ± 0.16	1.37 ± 0.58	1.00 ± 0.25	1.93 ± 0.45	NS
	FGF R1	0.86 ± 0.07	0.89 ± 0.08	0.92 ± 0.07	0.86 ± 0.07	1.09 ± 0.12	1.00 ± 0.06	1.06 ± 0.08	NS
	FGF R2B	0.93 ± 0.12	0.83 ± 0.15	0.63 ± 0.06	0.71 ± 0.06	0.99 ± 0.20	1.00 ± 0.20	0.99 ± 0.17	NS
	FGF R2C	0.84 ± 0.13 ^a^	0.79 ± 0.08 ^a^	0.80 ± 0.09 ^a^	0.78 ± 0.07 ^a^	1.13 ± 0.14 ^a^	1.00 ± 0.08 ^a^	1.14 ± 0.09 ^a^	<0.05
	FGF R3	0.96 ± 0.13	0.70 ± 0.08	0.74 ± 0.08	0.82 ± 0.12	1.06 ± 0.17	1.00 ± 0.10	0.97 ± 0.10	NS
	NPY	0.97 ± 0.25	0.64 ± 0.14	0.92 ± 0.16	0.67 ± 0.10	0.68 ± 0.21	1.00 ± 0.19	0.78 ± 0.11	NS
	AGRP	0.90 ± 0.10	0.81 ± 0.09	0.96 ± 0.07	0.78 ± 0.13	1.01 ± 0.14	1.00 ± 0.13	1.04 ± 0.10	NS
	Y2R	0.88 ± 0.13	0.80 ± 0.09	0.97 ± 0.11	0.92 ± 0.09	0.72 ± 0.10	1.00 ± 0.07	0.97 ± 0.06	NS

The level of lysine deficiency is indicated by the number in diet’s name expressed in percentage of recommendation, except for control (P20). Values are expressed as means + SEM. Data that do not share the same letter are different at *p* < 0.05. Bold values are those significantly different from P20 group.

**Table 6 nutrients-15-00197-t006:** Absolute relative expression of mRNA as determined by RT-qPCR for threonine diets.

Organ	Diet	T15	T25	T40	T60	T75	P20	T100	Diet Effect
Liver	GK	0.38 ± 0.12	0.94 ± 0.15	1.25 ± 0.23	1.33 ± 0.34	1.40 ± 0.27	1.00 ± 0.23	1.50 ± 0.44	NS
	L-PK	1.04 ± 0.11	0.90 ± 0.18	1.91 ± 0.50	1.65 ± 0.36	1.21 ± 0.17	1.00 ± 0.18	1.03 ± 0.24	NS
	CD36	1.39 ± 0.15	1.10 ± 0.14	0.95 ± 0.13	1.23 ± 0.13	1.27 ± 0.12	1.00 ± 0.11	1.02 ± 0.09	NS
	CPT1a	1.10 ± 0.15 ^a^	1.11 ± 0.14 ^a^	1.84 ± 0.35 ^ab^	**2.37 ± 0.51 ^b^**	2.19 ± 0.19 ^ab^	1.00 ± 0.26 ^a^	1.74 ± 0.26 ^ab^	<0.01
	ACOX	0.73 ± 0.06 ^a^	0.76 ± 0.13 ^a^	1.24 ± 0.23 ^abc^	1.83 ± 0.41 ^bc^	**1.94 ± 0.24 ^c^**	1.00 ± 0.12 ^ab^	1.22 ± 0.15 ^abc^	<0.001
	ACCa	1.28 ± 0.17	1.04 ± 0.13	1.23 ± 0.18	1.77 ± 0.33	1.32 ± 0.14	1.00 ± 0.11	1.69 ± 0.37	NS
	FAS	3.01 ± 0.55 ^b^	2.15 ± 0.44 ^ab^	2.29 ± 0.41 ^ab^	1.98 ± 0.45 ^ab^	1.31 ± 0.21 ^ab^	1.00 ± 0.10 ^ab^	2.20 ± 0.51 ^ab^	<0.05
	MTTP	1.03 ± 0.07	0.91 ± 0.10	1.00 ± 0.10	1.35 ± 0.17	1.04 ± 0.17	1.00 ± 0.10	1.12 ± 0.07	NS
	FGF21	**17.23 ± 6.40 ^cde^**	**35.05 ± 14.42 ^e^**	**13.81 ± 2.97 ^de^**	1.80 ± 0.35 ^ab^	5.37 ± 3.18 ^abc^	1.00 ± 0.39 ^a^	**4.36 ± 1.28 ^bcd^**	<0.001
Muscle	ACOX	0.91 ± 0.14	1.06 ± 0.13	0.90 ± 0.08	0.87 ± 0.08	0.90 ± 0.06	1.00 ± 0.14	1.08 ± 0.09	NS
	CD36	1.59 ± 0.20 ^b^	1.41 ± 0.19 ^b^	0.78 ± 0.09 ^a^	0.75 ± 0.08 ^a^	0.69 ± 0.08 ^a^	1.00 ± 0.14 ^ab^	1.19 ± 0.16 ^ab^	<0.001
	CPT1b	1.08 ± 0.16	1.58 ± 0.33	0.85 ± 0.06	1.02 ± 0.11	0.96 ± 0.06	1.00 ± 0.15	1.19 ± 0.19	NS
EAT	ACCa	0.79 ± 0.15	1.22 ± 0.34	0.72 ± 0.07	0.71 ± 0.10	0.48 ± 0.08	1.00 ± 0.26	0.71 ± 0.11	NS
	FAS	0.70 ± 0.13	1.08 ± 0.27	0.71 ± 0.08	0.75 ± 0.10	0.46 ± 0.10	1.00 ± 0.22	0.71 ± 0.10	NS
	CD36	1.05 ± 0.26	1.03 ± 0.31	0.75 ± 0.12	0.61 ± 0.08	0.57 ± 0.06	1.00 ± 0.20	0.71 ± 0.19	NS
	UCP1	**5.48 ± 1.48 ^c^**	5.24 ± 2.05 ^bc^	3.40 ± 1.32 ^abc^	1.03 ± 0.46 ^abc^	1.43 ± 1.02 ^abc^	1.00 ± 0.25 ^ab^	0.68 ± 0.12 ^a^	<0.001
	UCP2	1.24 ± 0.41	1.22 ± 0.35	0.62 ± 0.06	0.77 ± 0.09	0.63 ± 0.05	1.00 ± 0.17	0.68 ± 0.13	NS
	UCP3	0.35 ± 0.08 ^a^	0.98 ± 0.29 ^b^	0.73 ± 0.08 ^ab^	0.58 ± 0.09 ^ab^	0.42 ± 0.05 ^ab^	1.00 ± 0.35 ^ab^	0.57 ± 0.09 ^ab^	<0.05
Hypothalamus	FGF21	0.71 ± 0.20	1.20 ± 0.29	0.92 ± 0.20	1.45 ± 0.34	1.34 ± 0.30	1.00 ± 0.22	1.29 ± 0.24	NS
	FGF R1	0.83 ± 0.09	1.01 ± 0.08	1.09 ± 0.14	1.18 ± 0.16	1.13 ± 0.13	1.00 ± 0.12	1.08 ± 0.08	NS
	FGF R2B	0.78 ± 0.06	0.99 ± 0.09	1.00 ± 0.16	1.08 ± 0.14	1.12 ± 0.12	1.00 ± 0.12	0.98 ± 0.03	NS
	FGF R2C	0.75 ± 0.07	0.97 ± 0.11	0.97 ± 0.14	1.06 ± 0.15	1.07 ± 0.12	1.00 ± 0.13	0.94 ± 0.02	NS
	FGF R3	0.96 ± 0.06	1.10 ± 0.05	1.08 ± 0.10	1.24 ± 0.12	1.24 ± 0.13	1.00 ± 0.08	1.15 ± 0.05	NS
	NPY	**2.13 ± 0.21 ^c^**	**1.69 ± 0.14 ^bc^**	1.31 ± 0.09 ^ab^	1.10 ± 0.11 ^ab^	1.25 ± 0.21 ^ab^	1.00 ± 0.06 ^a^	0.99 ± 0.06 ^a^	<0.001
	AGRP	**2.73 ± 0.26 ^b^**	**2.23 ± 0.18 ^b^**	1.42 ± 0.18 ^a^	1.29 ± 0.16 ^a^	1.27 ± 0.10 ^a^	1.00 ± 0.10 ^a^	1.15 ± 0.06 ^a^	<0.001
	Y2R	0.78 ± 0.10	1.04 ± 0.12	1.10 ± 0.19	1.18 ± 0.18	1.31 ± 0.26	1.00 ± 0.14	1.16 ± 0.12	NS
	POMC	**0.29 ± 0.04 ^a^**	**0.45 ± 0.05 ^ab^**	0.87 ± 0.14 ^bc^	1.10 ± 0.12 ^c^	1.14 ± 0.14 ^c^	1.00 ± 0.15 ^c^	1.08 ± 0.08 ^c^	<0.001
	CART	0.88 ± 0.06	0.94 ± 0.05	1.01 ± 0.06	1.12 ± 0.14	1.24 ± 0.08	1.00 ± 0.04	1.08 ± 0.09	NS
	MOR	0.81 ± 0.06	1.16 ± 0.17	1.07 ± 0.17	1.23 ± 0.23	1.34 ± 0.16	1.00 ± 0.13	1.12 ± 0.11	NS
	MC3R	0.85 ± 0.11	1.08 ± 0.10	1.11 ± 0.18	1.16 ± 0.17	1.03 ± 0.15	1.00 ± 0.16	1.22 ± 0.16	NS
	MC4R	0.79 ± 0.10	0.92 ± 0.06	0.98 ± 0.15	1.01 ± 0.15	1.45 ± 0.25	1.00 ± 0.12	1.05 ± 0.09	NS
	CRF	1.05 ± 0.17	1.28 ± 0.10	1.36 ± 0.26	1.28 ± 0.25	1.20 ± 0.14	1.00 ± 0.14	1.07 ± 0.12	NS

The level of threonine deficiency is indicated by the number in diet’s name expressed in percentage of recommendation, except for control (P20). Values are expressed as means + SEM. Data that do not share the same letter are different at *p* < 0.05. Bold values are those significantly different from P20 group.

## Data Availability

Data presented in this study are available on request from the corresponding author.

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
