# Peer review of "Lysine or Threonine Deficiency Decreases Body Weight Gain in Growing Rats despite an Increase in Food Intake without Increasing Energy Expenditure in Response to FGF21"

_nutrients, 2022, doi:10.3390/nu15010197_

Round 1

Reviewer 1 Report

This is overall a nice study that certainly adds to our understanding of EAA. This study is executed and designed well.

In section 2.3 experimental design, author's mentioned that 56 animals were divided in to seven groups and in section 2.2 experimental diets and in tables and figures, there are 14 groups (7 with L diets and 7 with T diets including controls). Please explain number of animals in each group (for example: L15, n=8).

Minor errors have been noted in Figure 1 (c) spelling mistake on Y axis "daily". In Table A1. L100 column please represent the uniform decimal separator "." instead of ",".

Authors have extensively used definitive article "the" throughout the manuscript text, need to be removed in places where not required.

Author Response

  • This is overall a nice study that certainly adds to our understanding of EAA. This study is executed and designed well.

Thank you.

  • In section 2.3 experimental design, author's mentioned that 56 animals were divided in to seven groups and in section 2.2 experimental diets and in tables and figures, there are 14 groups (7 with L diets and 7 with T diets including controls). Please explain number of animals in each group (for example: L15, n=8).

We have added a sentence at the beginning of part “2.3.” to precise the numbers of animals. Indeed, 56 rats (8 per diet with 7 diets) were used for each amino acid, so in total 112 animals were used.

  • Minor errors have been noted in Figure 1 (c) spelling mistake on Y axis "daily". In Table A1. L100 column please represent the uniform decimal separator "." instead of ",".

We have corrected Figure 1 (c) as Table A1 for these mistakes.

  • Authors have extensively used definitive article "the" throughout the manuscript text, need to be removed in places where not required.

Thank you for this remark. We have checked the manuscript and taken it off as much as possible.

Reviewer 2 Report

1.      The title of the manuscript should be revised

2.      The article as well as the title have been drafted without a clear rationale

3.      Abstract needs bit of attention and should cover the theme of whole manuscript

4.      A conclusive line should also be added at the end of the abstract.

5.      Spelling errors were observed in some instants

6.      Grammatical errors at several places

7.      Summarize recent research related to the topic

8.      Highlight gaps in current understanding or conflicts in current knowledge

9.      Establish the originality of the research aims by demonstrating the need for investigations in the topic area

10.  Give a clear idea of the target readership, why the research was carried out and the novelty and topicality of the manuscript

11.  Line 44 to 46. Too much lengthy sentences and doesn’t make sense. Rewrite

12.  Graphs should elaborate the clear results. Recheck it

13.  Discussion should be improved by discussing the different trends.

14.  In the discussion part, also mention the body mechanism, and how EAAs are helpful.

15.  Only use scientific words throughout the manuscript

16.  Conclusion should be revised for improving the reader's understanding

17. The plagiarism (40%) is detected and one source 32%. It should be less than 15% overall

18.  In a single source it should be less than 5%

Author Response

  1. The title of the manuscript should be revised

We propose as a title:

Lysine or threonine deficiency decreases body weight gain in growing rats despite an increase of food intake and without increasing energy expenditure in response to FGF21.

  1. The article as well as the title have been drafted without a clear rationale

The objective of the research work was indicated in the first sentence of the abstract (line 16 and 17), at the end of the introduction (lines 76 to 78), at at the beginning of the discussion (lines 381 to 383). This study aims to evaluate the effects of a strictly essential amino acid (lysine or threonine; EAA) deficiency on energy metabolism in growing rats. In the introduction and the discussion, we developed the literature on the subject and we discussed our results according to the literature. Thus, for us the rational hope it is now more appropriate. Of the manuscript is clear but maybe we did not understand the comment of the reviewer. We changed the tittle adding the results and we

  1. Abstract needs bit of attention and should cover the theme of whole manuscript

We modified the abstract but it is difficult to add all the results in the abstract which should be less than 200 words. The main results according to the objective are indicated in the abstract.

  1. A conclusive line should also be added at the end of the abstract.

We add a conclusive sentence at the end of the abstract

  1. Spelling errors were observed in some instants

We have checked the manuscript again to improve the spelling and grammatical.

  1. Grammatical errors at several places

We have checked the manuscript again to improve the spelling and grammatical.

  1. Summarize recent research related to the topic.

The recent work and publications on the subject were summarized in the introduction. Are there any scientific papers missing?

  1. Highlight gaps in current understanding or conflicts in current knowledge

In the introduction, at the end of each paragraph, we have indicated the current knowledge and what remains to do. Similarly, in the discussion, we have discussed our results according to the literature, by indicating what agrees or conflicts with our results.

  1. Establish the originality of the research aims by demonstrating the need for investigations in the topic area

We would like to make the changes requested by the reviewer but the comments are too general and we do not know how we can improve the manuscript. In the introduction, we presented the need for investigations in the topic area according to the literature review.

We have rewritten the conclusion to highlight the originality of our work

  1. Give a clear idea of the target readership, why the research was carried out and the novelty and topicality of the manuscript

We add at the end of the introduction a sentence at the end of the introduction and we hope that this could help the target readership to understand our work.

  1. Line 44 to 46. Too much lengthy sentences and doesn’t make sense. Rewrite

These sentences have been changed. Thanks for your remark.

  1. Graphs should elaborate the clear results. Recheck it

We would like to make the changes requested by the reviewer but the comments are too general and we do not know how we can improve the graphs.

  1. Discussion should be improved by discussing the different trends.

We would like to make the changes requested by the reviewer but the comments are too general and we do not know how we can improve the discussion

  1. In the discussion part, also mention the body mechanism, and how EAAs are helpful.

We would like to make the changes requested by the reviewer but the comments are too general and we do not know how we can improve the discussion. Along the discussion, we propose some mechanisms for EAA effects on body weight and food intake, including FGF21, neuropeptides, energy expenditure. We have avoided being too speculative.

  1. Only use scientific words throughout the manuscript

We would have liked to make the changes requested by the reviewer but the comments are too general and we do not know how we can improve. Where did we use no scientific words?

  1. Conclusion should be revised for improving the reader's understanding

We rewrite the conclusion to highlight the originality of our work

  1. The plagiarism (40%) is detected and one source 32%. It should be less than 15% overall

The score of plagiarism is really high as this work was a part of a PhD work, which includes the draft of the manuscript. Indeed, it is normal that plagiarism was high. Moreover, there is also a similarity with our previous publication on low-protein due to the topic and the design of the studies.

  1. In a single source it should be less than 5%

Except for our previous work, no other source was detected with a plagiarism score more than <1%.
